# Microbial Richness of Marine Biofilms Revealed by Sequencing Full-Length 16S rRNA Genes

**DOI:** 10.3390/genes13061050

**Published:** 2022-06-12

**Authors:** Shougang Wang, Xiaoyan Su, Han Cui, Meng Wang, Xiaoli Hu, Wei Ding, Weipeng Zhang

**Affiliations:** College of Marine Life Sciences, Ocean University of China, Qingdao 266005, China; wangshougang@stu.ouc.edu.cn (S.W.); suxiaoyan@stu.ouc.edu.cn (X.S.); cuihan@stu.ouc.edu.cn (H.C.); wangmeng@stu.ouc.edu.cn (M.W.); hxl707@ouc.edu.cn (X.H.); dingwei@ouc.edu.cn (W.D.)

**Keywords:** PacBio sequencing, Illumina sequencing, biofilm, microbial richness, 16S rRNA gene

## Abstract

Marine biofilms are a collective of microbes that can grow on many different surfaces immersed in marine environments. Estimating the microbial richness and specificity of a marine biofilm community is a challenging task due to the high complexity in comparison with seawater. Here, we compared the resolution of full-length 16S rRNA gene sequencing technique of a PacBio platform for microbe identification in marine biofilms with the results of partial 16S rRNA gene sequencing of traditional Illumina PE250 platform. At the same time, the microbial richness, diversity, and composition of adjacent seawater communities in the same batch of samples were analyzed. Both techniques revealed higher species richness, as reflected by the Chao1 index, in the biofilms than that in the seawater communities. Moreover, compared with Illumina sequencing, PacBio sequencing detected more specific species for biofilms and less specific species for seawater. Members of *Vibrio*, *Arcobacter*, *Photobacterium*, *Pseudoalteromonas*, and *Thalassomonas* were significantly enriched in the biofilms, which is consistent with the previous understanding of species adapted to a surface-associated lifestyle and validates the taxonomic analyses in the current study. To conclude, the full-length sequencing of 16S rRNA genes has probably a stronger ability to analyze more complex microbial communities, such as marine biofilms, the species richness of which has probably been under-estimated in previous studies.

## 1. Introduction

Marine microbial communities in the ocean account for more than 80% of biomass on earth and have crucial ecological roles [1,2]. For example, marine microorganisms are the most important primary producers in the ocean; they convert carbon dioxide into organic matter through photosynthesis and chemosynthesis and support the whole marine ecosystem through food chains and food webs [2,3]. At the same time, microbes are considered to be the main decomposers in ecosystems, which reconvert complex organic matter into relatively simple organics [4,5]. For these reasons, there is a strong motivation to study the diversity of marine microorganisms. The methods originally used to study microbial diversity were mostly direct observation under the microscope and plate counting [6], but assessing marine microbial diversity is a challenging task due to their sheer variety and resistance to cultivation [7].

In the marine environment, a biofilm can be defined as a community of microbes attached to the surface of a solid material, such as the surface of rocks, floating artificial panels, and animal guts [8,9,10]. Zhang et al. collected biofilms from various substrates, including natural and artificial materials, in eight countries around the world and obtained 101 biofilm metagenomes using Illumina sequencing. The results indicated very high microbial richness in the biofilm communities, which contained microbes (more than 7300 species as revealed by the miTags extracted from the 101 biofilm metagenomes) with undetectable abundance in the seawater [11]. As such, biofilms hugely supplement marine microbial richness and represent bounteous reservoirs for exploring marine microbial resources.

The emergence of DNA-sequencing technologies at the end of last century has made it possible to directly detect microbial diversity in natural environments, which has greatly benefited the field of microbial ecology [6]. So far, three main techniques have been used to sequence microbial diversity. The Illumina platforms that are used to sequence environmental metagenomes are effective means of characterizing both taxonomic and functional diversity; 16S rRNA genes extracted from metagenomes are defined as miTags, which have the advantage of more accurately reflecting real community richness [12], and as mentioned above, this method has been used in analyzing marine biofilms [11]. Illumina platforms can also be employed to detect target variable regions in the highly conserved 16S rRNA genes after PCR using universal primers. The V3–V4 and V4–V5 variable regions are the most common sequences used in the study of natural environment samples [13,14,15]. The V2–V3 variable regions have higher resolution for lower taxonomic levels (i.e., genus and species level) [16], and the V6–V8 variable regions have better recognition ability for lower abundance taxa [17], whereas the V1–V3 [18,19], V3–V5 [20], V4–V6 [21], and V5–V9 [22] regions are also used for profiling microbial community compositions under specific environments. In third-generation sequencing techniques, i.e., PacBio sequencing, the read length completely covers the full-length 16S rRNA genes, with an average length of 1.5 kb, providing more information on native DNA bases [23,24]. Studies have shown that PacBio sequencing can detect more operational taxonomic units (OTUs) with higher resolution at the genus and species levels and demonstrated it to be a useful way to profile marine microbial communities [25,26].

The major purpose of this study is to analyze the microbial richness, diversity, and composition of marine biofilms through sequencing full-length 16S rRNA genes and the V3–V4 regions; we also test whether PacBio sequencing is more effective in dealing with complex microbial communities, through comparing the results of biofilms with the results of seawater communities. After extracting DNA from the biofilm and seawater samples, 16S rRNA gene amplicon sequencing was conducted using PacBio and Illumina techniques to sequence the full-length and the V3–V4 region of 16S rRNA genes, respectively. Then, the microbial richness, diversity, and community structure were comparatively analyzed.

## 2. Methods

### 2.1. Biofilm and Seawater Sampling

Following the steps described in our previous study [11], biofilms were developed on polystyrene Petri dishes with a diameter of 9 cm. The Petri dishes were placed in a 20 × 30 cm nylon net bag and then put into the subtidal zone along the coast of Qingdao (120.145, 39.915). After they were immersed in seawater for two weeks, the panels were transferred to the laboratory and subjected to sample preparation immediately. Briefly, the biofilms attached to the plate were scraped off with a cotton swab and put into 50 mL centrifuge tubes with artificial seawater to vibrate for 15 min. The seawater samples were filtered with 0.22-micron filter membranes, which were then cut, crushed, and put into 50 mL centrifuge tubes filled with artificial seawater to vibrate for 15 min. Finally, the centrifuge tubes were put into a centrifuge (6000 rpm) to obtain bacterial cells for DNA extraction.

### 2.2. DNA Extraction, PCR, and Sequencing

DNA from the biofilm and seawater samples was extracted using a One-Step-Lysis Bacteria DNA Kit (Nobelab Biotechnology, Beijing, China). For the PacBio sequencing, a 50 μL amplification system was prepared, which consisted of 25 μL of Prime Star Max premix (R045, Takara, Beijing, China), 2.5 μL of 27F primers (5′-AGAGTTTGATCMTGGCTCAG-3′) and 2.5 μL of 1492R primers (5′-GGTTACCTTGTTACGACTT-3′) with six different barcodes, 1 μL of DNA template, and distilled water to the remaining volume. The PCR amplification conditions included a denaturation step at 95 °C for 5 min, 30 cycles of amplification (95 °C for 30 s, 57 °C for 30 s, and 72 °C for 2 min), and a final extending step at 72 °C for 5 min. PCR products were examined using agarose gel electrophoresis and a Qubit assay before being processed on a PacBio RS II sequencing platform by Novogene (Beijing, China). The sequencing depth was 100,000 reads per sample. For Illumina sequencing, the same PCR procedures were conducted using the 341F primers (5′-CCTACGGGNGGCWGCAG-3′) and 805R primers (5′-GACTACHVGGGTATCTAATCC-3′) with six different pairs of barcodes. The PCR products were sequenced on an Illumina PE250 platform by Novogene (Beijing, China), with a sequencing depth of 100,000 pair-ended reads per sample.

### 2.3. Data Correction and Filtration

The offline data from PacBio sequencing were distinguished and segregated based on the different barcode sequences added in the PCR process. The raw sequences were saved in bam format without correction. Then, circular consensus sequencing (CCS) (SMRT link v7.0) was used to correct the sequences. The correction parameters were the following: CCS = 3, minimum accuracy = 0.99. The sequence lengths ranged between 1340 bp and 1640 bp. The files were stored in fastq and fasta formats. Sequences containing consecutive identical base numbers of more than 8 were filtered out using cutadapt [27] to generate the final valid data (clean reads) for analysis. Pair-ended Illumina reads were assigned to samples based on their unique barcodes and truncated by cutting off the barcode and primer sequences. FLASH (v1.2.7) software was used to splice the overlapping read pairs [28]. QIIME2 [29] was used to filter out sequences with undetected bases (represented by N) or low-quality bases (score < 75%) and, finally, the chimeric sequences generated by PCR errors. The reads obtained after the above processing steps were considered the final valid data (clean reads).

### 2.4. Data Analysis

Parallel-META3 (v3.5) [30] was employed to classify the PacBio and Illumina clean reads using a similarity cutoff of 97%. The mapped OTU IDs in the Parallel-META3 database file were used as the first column in the construction of an OTU table that integrated both PacBio and Illumina reads. OTUs that contained less than two reads (total number) were removed from further analysis. The relative abundance of a given OTU was defined as the read number it contained. The relative abundance of a given phylum or genus was indicated by summarizing the read numbers contained by all the OTUs that were classified into the same phylum or genus. The multiple_rarefactions.py, α_diversity.py, and collate_α.py script documented in QIIME2 [29] were used to draw rarefaction curves at intervals of 1000 sequences with 10 replicated calculations. α-diversities based on the Shannon diversity (H index) and Chao1 richness were calculated. β-diversity, as indicated by the principal coordinate analysis (PCoA), was performed using PAST (v4.03) [31] and checked using ASAP2 [32] after transforming the OTU compositions into a Bray–Curtis distance matrix. Finally, a statistical analysis was performed using a two-tailed Student’s *t*-test after confirming that the data were normally distributed.

## 3. Results

### 3.1. Richness, Diversity, and Rarefaction Curves

PacBio RS II was used to sequence the full-length 16S rRNA genes of six samples, including three biofilm and three seawater samples, which generated 140,859, 191,358, 178,835, 154,650, 130,362, and 126,329 clean reads for Biofilm 1, 2, and 3 and Seawater 1, 2, and 3, respectively (Table 1). Illumina PE250 was used to sequence the 16S rRNA genes (V3–V4 region) of the same samples, which generated 97,651, 97,487, 99,715, 97,732, 96,248, and 104,682 respective clean reads (Table 1). Hereafter, the PacBio data were designated PB-Biofilms and PB-Seawater, while the Illumina data were designated Illu-Biofilms and Illu-Seawater. The OTU table comprising all the datasets was constructed by mapping the sequences to a reference OTU database documented in Parallel-META3 [30]. The detected OTUs ranged from 1203 to 3937 in the 12 data files. Overall, more OTUs were detected in the PacBio-derived biofilm data (3851 OTUs on average), which detected 34.306% more OTUs in comparison with the Illumina-derived biofilm data (2867 OTUs on average) (Table 1). However, for the seawater community, Illumina sequencing detected more OTUs than the PacBio sequencing did (1711 vs. 1399) (Table 1).

Rarefaction curves were drawn based on the OTU table to examine whether the sequencing depth was sufficient to reflect species richness and to compare diversities. Chao1 richness as well as Shannon diversity (H index) were calculated based on the PacBio and Illumina datasets, followed by rarefaction curve construction. In all the examined samples, the curves reached saturation, suggesting that the sequencing depth was enough to cover most species (Figure 1). In most of the comparisons, α-diversity values were higher for the biofilms than for the seawater communities; interestingly, however, Illumina sequencing generated higher Shannon diversity for the seawater than for the biofilms (Figure 1), suggesting that microbial community evenness is influenced by sequencing type. On the other hand, the biofilms’ Chao1 and Shannon values generated by PacBio sequencing were higher than those generated by Illumina sequencing; whereas, for the seawater samples, Illumina sequencing revealed similar or even higher microbial diversity than PacBio sequencing did (Figure 1).

### 3.2. PCoA

To reveal similarities between the datasets generated by different techniques, as well as between the biofilm and seawater samples, a β-diversity analysis was performed using PCoA. Based on Bray–Curtis distances, which consider both the identity and abundance of the OTUs, the six biofilm datasets were separated from the six seawater datasets along the PC1 axis, which explained 34.026% of the difference (Figure 2). Moreover, dissimilarities between the PB-Biofilms and Illu-Biofilms were observed, as was the case for the PB-Seawater and Illu-Seawater communities (Figure 2).

### 3.3. Venn Analysis

To further visualize the dissimilarity between biofilm and seawater samples, a Venn analysis was conducted. OTUs from the three PB-Biofilms were pooled to compare with the three combined PB-Seawater samples. As shown by the results, 2003 (35.75%) OTUs were shared by the biofilm and seawater groups, while 2809 (50.13%) and 791 OTUs (14.12%) were specific to the biofilm group and seawater group, respectively (Figure 3A). In parallel, OTUs from the three Illu-Biofilms were pooled to compare with the combined Illu-Seawater samples, which revealed that 1855 (37.82%) OTUs were shared by the biofilm and seawater groups, while 2167 (44.18%) and 883 OTUs (18.00%) were specific to the biofilm group and seawater group, respectively (Figure 3B). Taken together, PacBio sequencing revealed the presence of more specific OTUs for biofilms and less specific species for seawater communities.

### 3.4. Taxonomic Composition

To further study the dissimilarity between the biofilm and seawater samples, the taxonomic structure was profiled at the phylum level using the OTU table. Based on the PacBio datasets, 52 phyla were identified in the biofilms, whereas 47 phyla were identified in the seawater samples. The biofilms were primarily composed of Proteobacteria (70.958–79.604%), Bacteroides (8.979–15.448%), and Cyanobacteria (3.526–8.301%), whereas the seawater samples were predominantly composed of Proteobacteria (59.787–60.258%), Cyanobacteria (10.993–16.527%), Bacterioidetes (12.263–15.171%), and Actinobacteria (7.238–9.204%) (Figure 4A). PacBio sequencing showed that 44 phyla were shared among the biofilms and seawater communities, while AncK6, FCPU426, WPS_2, Caldithrix, WS5, Hyd24_12, KSB3, and Armatimonadetes were only present in the biofilms, and Crenarchaeota, Deferribacteres, and WS1 were only present in the seawater communities (Figure 4A). In parallel, Illumina sequencing revealed the presence of 57 and 52 phyla in the biofilm and seawater samples, respectively. Proteobacteria (84.310–94.900%) and Bacteroidetes (2.230–6.807%) made up most of the read sequences for the biofilms, whereas Proteobacteria (45.451–60.317%), Actinobacteria (15.582–19.125%), Cyanobacteria (5.992–13.110%), and Bacteroidetes (9.567–12.814%) were responsible for most of the read sequences for the seawater communities (Figure 4B). Illumina sequencing showed that 48 phyla were present in both biofilms and seawater, while Anck6, LD1, WWE1, WS2, OP1, KSB3, Poribacteria, Fcpu426, and Caldiserica were specific to the biofilms, and TPD_58 sar406, AC1, and Thermotogae were specific to the seawater communities (Figure 4B). Taken together, the results of both sequencing techniques indicated that there were more phyla in the biofilms than the seawater communities, and the biofilms possessed more specific phyla than the seawater communities. For both the biofilms and seawater communities, Illumina sequencing indicated the presence of more phyla than PacBio sequencing.

OTUs were then classified to the genus level. Based on the PacBio datasets, 678 genera were identified in the biofilms and were dominated by *Vibrio*, which matched about half of the sequences (40.221–49.222%), followed by *Arcobacter* (4.110–7.097%) and *Photobacterium* (3.177–5.942%). According to the PacBio datasets, only 526 genera appeared in the seawater samples, including *Candidatus*_Portiera (24.113–29.628%), *Candidatus*_Pelagibacter (11.571–18.514%), *Candidatus*_Actinomarina (6.452–8.675%), and *Synechococcus* (3.304–5.345%) (Figure 5A). PacBio sequencing showed that there were 225 genera specific for the biofilms and 73 genera specific for the seawater communities. In parallel, the Illumina datasets revealed 605 genera in the biofilms, which were dominated by *Vibrio* (52.511–66.727%), followed by *Catenoccus* (7.761–8.412%), *Pseudoalteromonas* (4.337–7.389%), and *Photobacterium* (4.843–7.138%) (Figure 5B). According to the Illumina datasets, there were 537 genera in the seawater samples, including *Candidatus*_Portiera (14.403–18.685%), *Candidatus*_Actinomarina (12.379–15.769%), *Roseovarius* (8.350–14.460%), *Synechococcus* (3.284–11.054%), and NS4 (2.490–7.841%). Illumina sequencing showed that 174 genera were specific to the biofilms, while 105 genera were specific to seawater. Taken together, the results of both sequencing techniques indicated that the biofilms possessed more genera as well as more specific genera than the seawater communities. Interestingly, PacBio sequencing of the biofilms revealed more genera than Illumina sequencing did, whereas PacBio sequencing of the seawater communities revealed fewer genera than Illumina sequencing did.

We further performed a statistical analysis to identify genera that were significantly enriched in the biofilms in comparison with the seawater communities. According to the PacBio datasets, *Vibrio*, *Thalassomonas*, *Pseudoalteromonas*, *Photobacterium*, *Arcobacter*, and *Aquibacter* were significantly (*p*-value <0.01) enriched in the biofilms (Figure 6A). In particular, the PacBio datasets suggested that *Vibrio* species were enriched by more than 100-fold in the biofilms (53.64% in biofilms compared to 0.52% in the seawater communities) (Figure 6A). According to the Illumina datasets, *Vibrio*, *Pseudoalteromonas*, and *Photobacterium* were also significantly enriched in the biofilms (Figure 6B). In addition, the Illumina datasets revealed *Catenococcus* to be a biofilm-enriched genus instead of *Arcobacter* (not detected) and *Aquibacter* (enriched, but not significantly) (Figure 6B).

## 4. Discussion

In the present study, we performed comparative analyses of the microbial richness, diversity, and composition of biofilms and adjacent seawater samples, based on 16S rRNA gene sequences generated by PacBio and Illumina sequencing. The results revealed higher microbial richness in the biofilms than the seawater communities. For the biofilms, PacBio sequencing revealed more species than Illumina sequencing did. The taxonomy was further analyzed to indicate biofilm specificity as well as the advantages of the PacBio sequencing for profiling complex microbial compositions.

The higher richness of the marine biofilms compared with the seawater communities observed in the present study and several previous studies [11,33] can be attributed to biofilms facilitating the proliferation of many species that are very rare in seawater. Such proliferation is probably related to the special biofilm architecture, which provides suitable physiological and chemical conditions, as well as ecological advantages, for these microbes. It is well known that biofilm matrixes formed by the polymerization of a large number of metabolites can improve the utilization efficiency of internal resources and signal transduction [34,35] and also improve the tolerance of bacteria in the community to external stresses, including metal ions and oxidative stresses [33,36]. Moreover, the uneven distribution of polymers and microbes within a biofilm facilitates the formation of many microenvironments with different oxygen gradients. The formation of such oxygen gradients allows the coexistence of microbes with different respiration abilities, i.e., preferences for different electron acceptors [37,38]. Actually, microbes can cooperate with each other to promote electron transduction and energy generation, which are important driving forces of biofilm development in environments with poor carbon and energy sources [39,40]. In line with these views, both PacBio and Illumina sequencing approaches identified *Vibrio*, *Catenococcus*, and *Pseudoalteromonas* as dominant and specific biofilm inhabitants in the present study, suggesting that these genera have evolved surface-associated lifestyles. A biofilm-forming ability brings survival advantages, such as enabling *Vibrio cholerae* to survive protozoan grazing [41] and enhancing *Vibrio* virulence [42]. *Pseudoalteromonas* species also have a strong ability to form biofilms. Many members of *Pseudoalteromonas* secrete anti-metabolites and can form a protective film inside the biofilm to enhance the resistance of the whole community to antibiotics [43]. At the same time, several *Pseudoalteromonas* species have antibacterial abilities and can parasitize the hosts to form biofilms that help the host fight against pathogenic bacteria [43]. The detection of these microbes has also validated the taxonomic profiling results in the present study.

Compared with Illumina sequencing, PacBio sequencing detected more species as well as more specific species in the biofilms. These results are consistent with previous conclusions that PacBio has great advantages for categorization to the species level and is more accurate than techniques measuring some variable regions [25,26]. On the other hand, the PCoA showed substantial dissimilarity between the PacBio- and Illumina-sequenced biofilms, and only 18 of the top 30 genera in the PacBio-sequenced biofilms were recalled by Illumina sequencing. Such a substantial difference may have arisen because of the full-length sequencing of 16S RNA, which not only contains the sequences comprising the V3–V4 variable region, but also information on all other variable regions. This view is also supported by findings that show that 16S rRNA regions have different advantages for identifying distinct microbes, largely depending on their taxonomy. For example, the V1–V2 region can be used to accurately identify a large number of clinically important strains [44,45], whereas the V1–V3 variable region can reveal a large number of endemic microbial groups [46]. However, it is worth pointing out that the primers used for the full-length 16S rRNA amplification and PacBio sequencing (27F/1492R) fail to fully cover the conserved regions of archaea, whose identification often rely on the 23F/1492R primer pair [47,48], and this also contributes to sequencing technique-generated taxonomic differences. Moreover, considering the different microbial richness between biofilms and seawater communities, the advantages of different sequencing strategies are probably related to the complexity of the target microbial communities.

## 5. Conclusions

To conclude, this work has highlighted the complexity of marine biofilms with extremely high and specific microbial richness in comparison with free-living microbial communities in seawater. Considering that most published microbial community research has focused on seawater, future studies on biofilms should further change our understanding of the ecology and lifestyle of marine microbes. More importantly, we found that using PacBio sequencing can better reflect the profiles of complex microbial communities, such as biofilms. In addition, we believe that the development and application of novel DNA sequencing techniques will continuously refresh our understanding of the microbial richness in the global ocean, which might have been severely underestimated in global sampling projects performed in past years [49,50].

## Figures and Tables

**Figure 1 genes-13-01050-f001:**
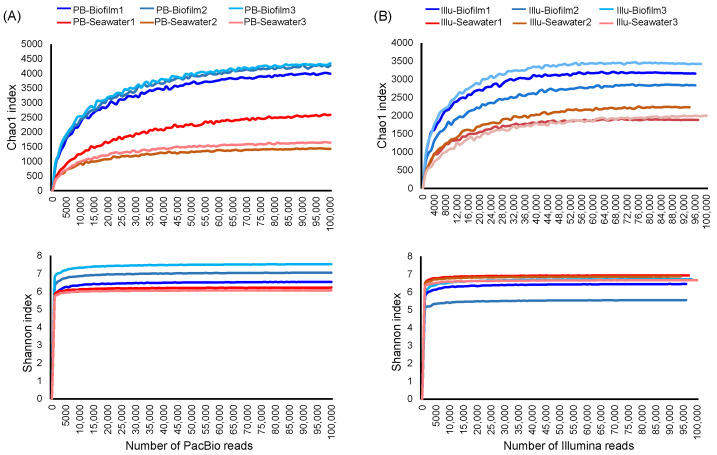
Rarefaction curves showing sequence number and α-diversity correlations. The three biofilm and three seawater samples are indicated by different colors. Chao1 richness and Shannon (H index) diversity were calculated based on the PacBio (**A**) and the Illumina datasets (**B**). For each sample, 100,000 sequences were included for analysis, and the average value of 10 permutations is shown.

**Figure 2 genes-13-01050-f002:**
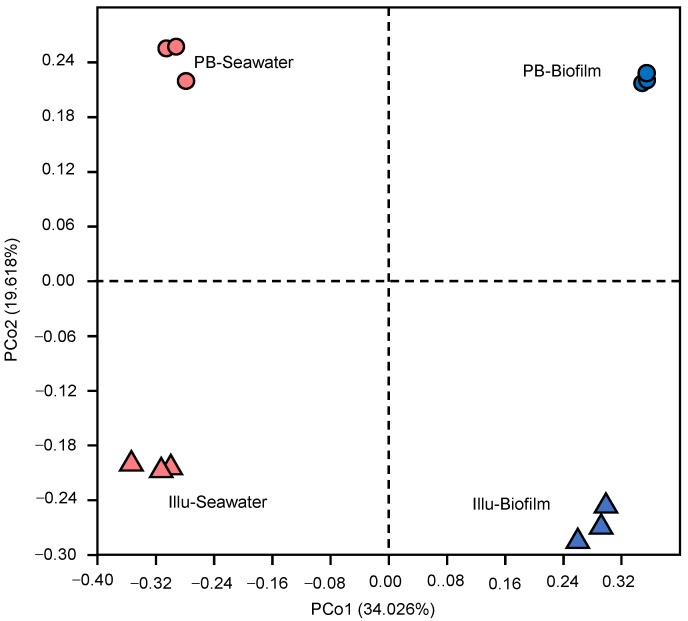
Principal coordinate analysis (PCoA) based on OTU compositions of biofilms and seawater communities. Bray–Curtis distances were used in the calculation. PB-Biofilm and PB-Seawater represent the three biofilm and three seawater samples sequenced by PacBio, respectively. Illu-Biofilm and Illu-Seawater represent the same samples sequenced by Illumina.

**Figure 3 genes-13-01050-f003:**
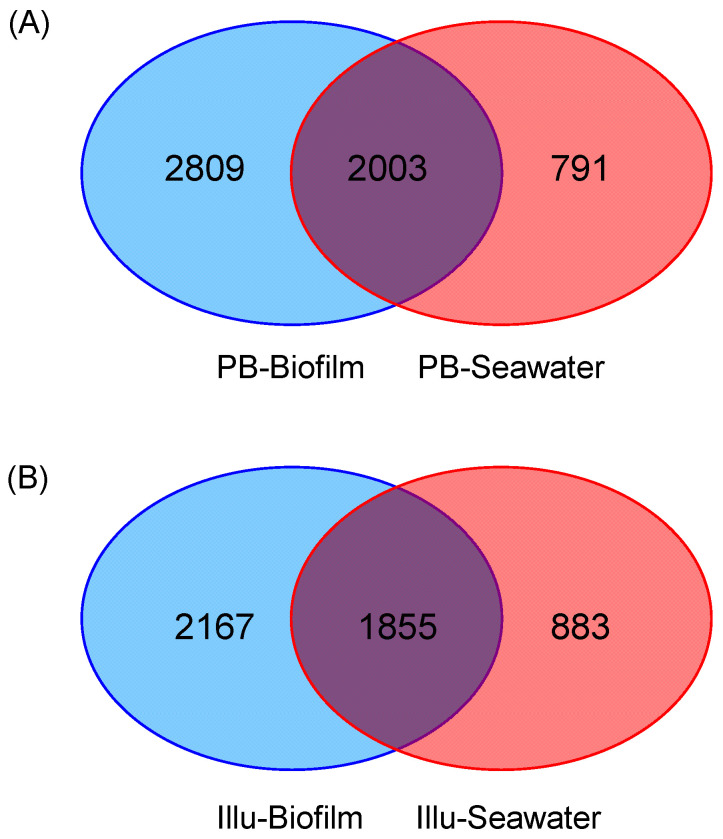
Venn diagrams showing the distribution of OTUs across biofilm and seawater samples. The three replicates were pooled together for analysis. (**A**) PacBio-sequenced samples. (**B**) Illumina-sequenced samples.

**Figure 4 genes-13-01050-f004:**
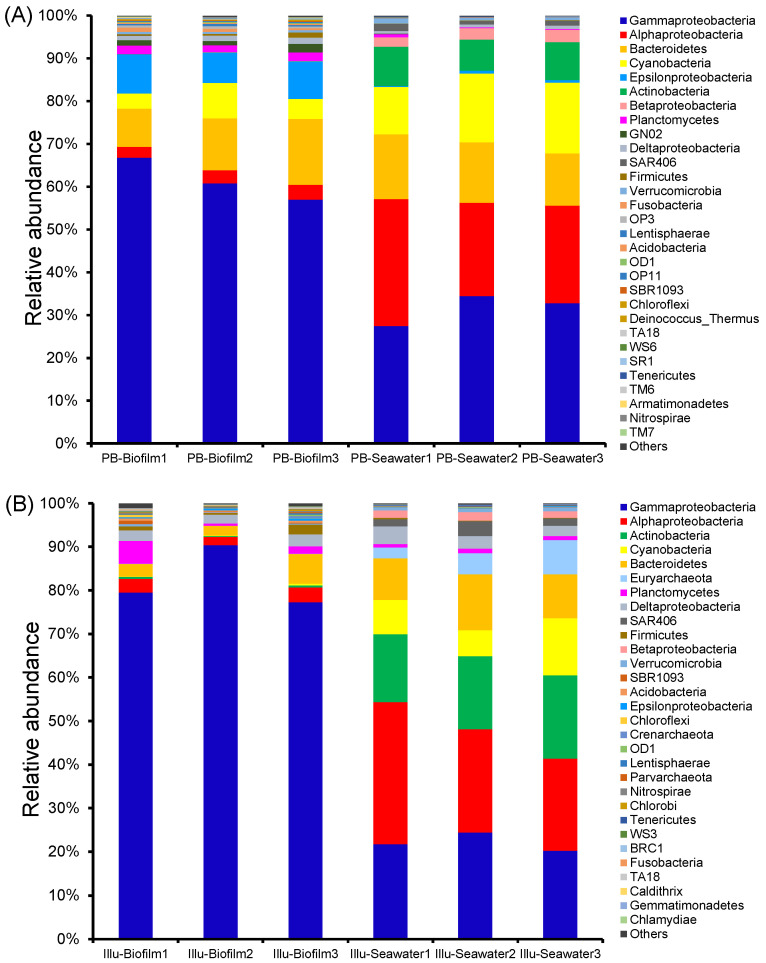
Phylum-level microbial community composition of the six samples. Proteobacteria were classified down to the class level. The 30 most abundant genera in terms of maximum relative abundance are shown, and the rest are merged into ‘Others’. (**A**) Biofilm and seawater microbial communities by PacBio sequencing. (**B**) Biofilm and seawater microbial communities by Illumina sequencing.

**Figure 5 genes-13-01050-f005:**
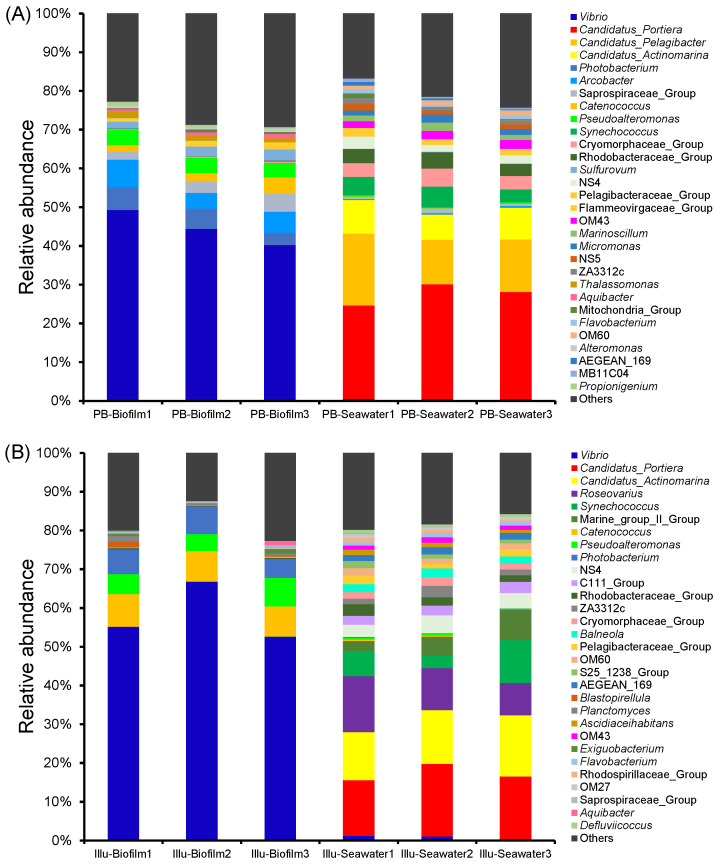
Genus-level taxonomic structure of biofilm and seawater samples. The top 30 abundant genera in terms of maximum relative abundance are shown, and the rest are merged into ‘Others’. (**A**) Biofilm and seawater microbial communities by PacBio sequencing. (**B**) Biofilm and seawater microbial communities by Illumina sequencing.

**Figure 6 genes-13-01050-f006:**
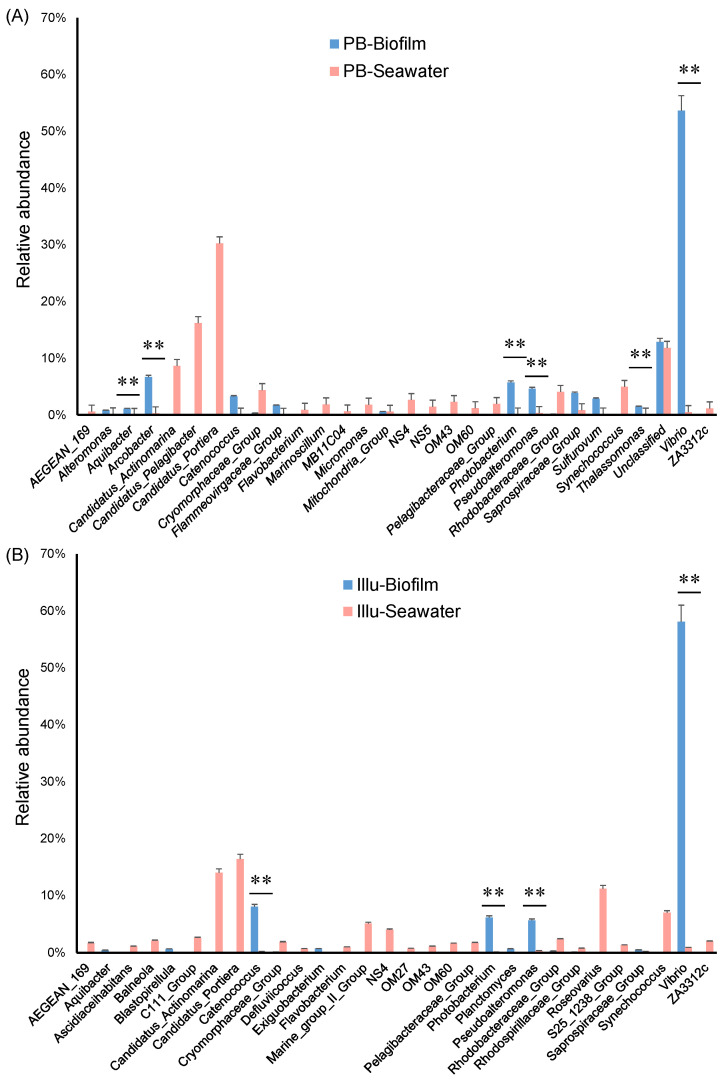
Statistical analysis of the 30 most abundant genera in biofilm and seawater samples. There were significant differences in genera between the biofilm and the seawater communities. ‘**’ represents a *p*-value < 0.01, calculated by two-tailed Student’s *t*-test. (**A**) Biofilm and seawater microbial communities by PacBio sequencing. (**B**) Biofilm and seawater microbial communities by Illumina sequencing.

**Table 1 genes-13-01050-t001:** Reads and OTUs information.

Sample	Clean Reads	OTU Number	Average OTU Number
PB-Biofilm1	140,859	3742	3851
PB-Biofilm2	191,358	3874	
PB-Biofilm3	178,835	3937	
PB-Seawater1	154,650	1690	1399
PB-Seawater2	130,362	1203	
PB-Seawater3	126,329	1305	
Illu-Biofilm1	97,651	2953	2867
Illu-Biofilm2	97,487	2462	
Illu-Biofilm3	99,715	3187	
Illu-Seawater1	97,732	1692	1711
Illu-Seawater2	96,248	1837	
Illu-Seawater3	104,682	1606	

## Data Availability

The data presented in this study are openly available on the NCBI database (Bioproject No. PRJNA816818).

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
