# Peer review of "Microbial Richness of Marine Biofilms Revealed by Sequencing Full-Length 16S rRNA Genes"

_genes, 2022, doi:10.3390/genes13061050_

Round 1

Reviewer 1 Report

Shougang Wang et al's manuscript "Microbial diversity of marine biofilms revealed by sequencing full-length 16S rRNA genes" explored and compared two major high throughput sequencing methods, namely PacBio platform for full-length 16S rRNA genes and Illumina-sequenced V3-V4 regions. To be honest, I think the significance of the content and the novelty of this manuscript are low. Numerous publications have discussed the topic. It is true the authors studied the marine biofilm this time but it is really not a matter. I can not approve the acceptance of this manuscript unless the authors can clearly demonstrate the significance of their work. Some other comments are shown below,

Line 7, the audience will appreciate it if the authors can define marine biofilm here.

Line 13, here "microbial diversity" is not accurate because ACE and Chao 1 both are abundance-based estimators of species richness rather than diversity. 

Line 15, "significantly" is ambiguous here. Do you mean statistically significant?

Line 17, Please confirm it is species richness or diversity.

Line 18, Please confirm it is richness or diversity because diversity includes richness and evenness. 

Line 24-25, Where are the keywords?

Line 28, "80% of life"? What do you mean? 80% species or biomass?

Line 64-73, The significance of this study object is low because previous studies have verified this. Please specify the significance of your study.

Line 147-151, No statistic analysis?

Line 154-156, How ACE and Chao1 been calculated? Could you please show the formulas? In Figure 1, the Y-axis of ACE and Chao1 seem odd because I have never seen so high numbers of ACE and Chao1.

Line 171, What is PCoA? Please refer to the full term in the main text for the first time.

Line 302, This is an odd way to show the number.

Author Response

Reviewer1

Shougang Wang et al's manuscript "Microbial diversity of marine biofilms revealed by sequencing full-length 16S rRNA genes" explored and compared two major high throughput sequencing methods, namely PacBio platform for full-length 16S rRNA genes and Illumina-sequenced V3-V4 regions. To be honest, I think the significance of the content and the novelty of this manuscript are low. Numerous publications have discussed the topic. It is true the authors studied the marine biofilm this time but it is really not a matter. I can not approve the acceptance of this manuscript unless the authors can clearly demonstrate the significance of their work. Some other comments are shown below,

Response: Thank you for your constructive comments on our study. According to your suggestions, we have corrected several deficiencies in the previous version. The detailed point-to-point responses are listed below.

Line 7, the audience will appreciate it if the authors can define marine biofilm here.

Response: We have added the definition of marine biofilms at the beginning of the abstract. Marine biofilms are a collective of microbes that can grow on many different surfaces immersed in marine environments.

Line 13, here "microbial diversity" is not accurate because ACE and Chao 1 both are abundance-based estimators of species richness rather than diversity.

Response: We have added the results of Shannon (H) index. According to the PacBio sequencing, biofilms showed higher Chao1 richness than the seawater communities did. Interestingly however, the Illumina sequencing generated higher Shannon diversity for the seawater than for the biofilms, suggesting that microbial community evenness is influenced by sequencing type.

Line 15, "significantly" is ambiguous here. Do you mean statistically significant?

Response: Yes. Members of many genera, including Vibrio, Arcobacter, Photobacterium, Pseudoalteromonas, and Thalassomonas, were significantly enriched (p-value < 0.01 and richness > 1%) in biofilms, consistent with our previous understanding of species adapted to a surface-associated lifestyle.

Line 17, Please confirm it is species richness or diversity.

Response: Chao1 richness, as well as Shannon diversity were calculated based on the Pac-Bio and Illumina sequencing.

Line 18, Please confirm it is richness or diversity because diversity includes richness and evenness. 

Response: We have performed the Shannon diversity calculation and revised the results. Chao1 richness, as well as Shannon diversity were calculated based on the Pac-Bio and Illumina sequencing.

Line 24-25, Where are the keywords?

Response: Added. “PacBio sequencing; Illumina sequencing; biofilm: microbial richness: 16S rRNA gene”

Line 28, "80% of life"? What do you mean? 80% species or biomass?

Response: 80% of biomass.

Line 64-73, The significance of this study object is low because previous studies have verified this. Please specify the significance of your study.

Response: The significance is that we have evaluated the correlation between sequencing type and community complexity. We found that, compared with Illumina sequencing, PacBio sequencing is more useful in dealing with complex microbial communities such as marine biofilms. However, when dealing with less complex communities, such as seawater communities, the advantage of PacBio sequencing cannot be well reflected.

Line 147-151, No statistic analysis?

Response: Statistical analysis was performed using two-tailed Student’s t-test after confirming that the data were normally distributed.

Line 154-156, How ACE and Chao1 been calculated? Could you please show the formulas? In Figure 1, the Y-axis of ACE and Chao1 seem odd because I have never seen so high numbers of ACE and Chao1.

Response: Chao1 = S + F1(F1 - 1) / (2 (F2 + 1)), where F1 is the number of singleton species and F2 the number of doubleton species.

Shannon Index (H) = ∑(i=1~s)pi*loge(pi), , where p is the proportion (n/N) of individuals of one particular species found (n) divided by the total number of individuals found (N), Σ is the sum of the calculations, and s is the number of species.

Line 171, What is PCoA? Please refer to the full term in the main text for the first time.

Response: Principal coordinate analysis (PCoA).

Line 302, This is an odd way to show the number.

Response: Revised according to the comments.

Reviewer 2 Report

Peer review for Genes

Microbial Diversity of Marine Biofilms Revealed by Sequencing Full-Length 16S rRNA Genes

Summary

This manuscript investigates microbial communities marine biofilms and marine water using PacBio sequencing of full-length 16S rRNA genes and compared the results with data for Illumina-sequenced V3–V4 regions. I have not found any real shortcoming in the methods used, or data produced, that would prevent acceptance of the present manuscript. The conclusions are sound, but alas not asserted through a robust argumentation since some of the supporting data is not shown. I consider that this study is of interest to the readership of Genes and would advise major revisions prior to resubmission. I will now summarize the main points that make the manuscript interpretations hard to follow or not consistent enough.

Abstract

It is necessary to better introduce the rationale of the study in one or two sentences. The authors don’t state their conclusions either.

Introduction

The introduction skips the big picture necessary using different methods for analysis microbial community.

The authors do not report how differ biofilms and water taken for analysis. 

Methods

The authors refer to previous studies without providing sufficient information to the reader who is not acquainted.

Results

NMDS data analysis could be performed to describe the results, as well as performed statistic comparisons of the genetic diversity estimates. And perform heat map for phyla representation and clustering samples with the Bray-Curtis distance metric for visualize obtained the data and compare the results.

Additional comment:

L. 24 missing keywords

L. 48 Also in the articles are presented the results on the V2-V3 region from natural ecosystems (for example, Bukin et al., 2019, Scientific Data)

L. 76 which steps are presented in the previous study. Сan you please explain. In previous study missing information about a coastal subtidal area of Qingdao (120.145, 39.915).

L. 102 Explain why Table 1 is presented in the article. This is already understandable.

L. 119 In the manuscript was used Parallel-META3 for taxonomy, but there is no discussion of why the authors used Parallel-META3 to describe diversity rather than SILVA. Currently, SILVA 138 is used as a reference program. Сan you please explain.

L. 302-325 In the discussion presents only the results obtained for the regions V1-V2, V3-V4, V4-V5, and V6-V8. These are not all possible regions for which there are now results of studies of microbial diversity.

L. 319-320 Identification of Archaea often rely on the 23F/1492R primer pair. Why it was not possible to describe the Archaea on this primer pair.

L. 336-342 missing informations

Author Response

Reviewer2

This manuscript investigates microbial communities marine biofilms and marine water using PacBio sequencing of full-length 16S rRNA genes and compared the results with data for Illumina-sequenced V3–V4 regions. I have not found any real shortcoming in the methods used, or data produced, that would prevent acceptance of the present manuscript. The conclusions are sound, but alas not asserted through a robust argumentation since some of the supporting data is not shown. I consider that this study is of interest to the readership of Genes and would advise major revisions prior to resubmission. I will now summarize the main points that make the manuscript interpretations hard to follow or not consistent enough.

Response: Thank you for your positive comments and valuable suggestions to improve the quality of our manuscript. According to your comments, we have made substantial changes to the manuscript. The detailed point-to-point responses are listed below.

Abstract

It is necessary to better introduce the rationale of the study in one or two sentences. The authors don’t state their conclusions either.

Response: To conclude, the full-length sequencing of 16S rRNA genes has probably a stronger ability to analyze more complex microbial communities such as marine biofilms.

Introduction

The introduction skips the big picture necessary using different methods for analysis microbial community.

Response: We have added more content in the Introduction part. The Illumina platforms that are used to sequence environmental metagenomes are effective means of characterizing both taxonomic and functional diversity; 16S rRNA genes extracted from metagenomes are defined as miTags, and this method has the advantage of more accurately reflecting real community richness and evenness [12]. Illumina platforms can also be employed to detect target variable regions in the highly conserved 16S rRNA genes after PCR using universal primers. The V3–V4 and V4–V5 variable regions are the most common sequences used in the study of natural environment samples [13-15]. In addition to the above variable areas, the V2-V3 variable regions have higher resolution for lower taxonomic levels (i.e., genus and species level) [16], the V6-V8 variable regions has better recognition ability for lower abundance taxa [17], whereas the V1-V3[18,19], V3-V5[20], V4-V6 [21], and V5-V9 [22] variable regions are also used for profiling microbial community compositions under specific environments.

The authors do not report how differ biofilms and water taken for analysis. 

Response: Zhang et al. collected biofilms from various substrates, including natural and artificial materials, in eight countries around the world and obtained 101 biofilm metagenomes using Illumina sequencing. The results indicated very high microbial richness and diversity in the biofilm communities, which contained microbes (more than 7300 species as revealed by the miTags extracted from the 101 biofilm metagenomes) with undetectable abundance in the seawater [11].

Methods

The authors refer to previous studies without providing sufficient information to the reader who is not acquainted.

Response: We have added more details in the Method part.

Results

NMDS data analysis could be performed to describe the results, as well as performed statistic comparisons of the genetic diversity estimates. And perform heat map for phyla representation and clustering samples with the Bray-Curtis distance metric for visualize obtained the data and compare the results.

Response: NMDS data analysis has been performed, and the result is similar to that generated by PCoA. Please see the attached pdf file.

Additional comment:

L24 missing keywords

Response: Added. ‘PacBio sequencing; Illumina sequencing; biofilm: microbial richness: 16S rRNA gene’.

L48 Also in the articles are presented the results on the V2-V3 region from natural ecosystems (for example, Bukin et al., 2019, Scientific Data)

Response: We have added the references about the 16S rRNA gene variable regions into the Introduction part.

L76 which steps are presented in the previous study. Сan you please explain. In previous study missing information about a coastal subtidal area of Qingdao (120.145, 39.915).

Response: Following the steps described in our previous study [11], biofilms were developed on polystyrene Petri dishes with a diameter of 9 cm. The Petri dishes were placed in a 20×30 cm nylon net bag, and then put into the subtidal zone along the coast of Qingdao (120.145, 39.915).

L102 Explain why Table 1 is presented in the article. This is already understandable.

Response: We have removed Table 1 in the revised manuscript.

L119 In the manuscript was used Parallel-META3 for taxonomy, but there is no discussion of why the authors used Parallel-META3 to describe diversity rather than SILVA. Currently, SILVA 138 is used as a reference program. Сan you please explain.

Response: Each independent database has certain limitations. Parallel-META 3, which integrates the GreenGenes, RDP, and SILVA databases, has the characteristics of small thread occupation and fast operation speed.

L302-325 In the discussion presents only the results obtained for the regions V1-V2, V3-V4, V4-V5, and V6-V8. These are not all possible regions for which there are now results of studies of microbial diversity.

Response: We have checked the literatures carefully and added these references on other variable areas into Discussion part.

L319-320 Identification of Archaeaoften rely on the 23F/1492R primer pair. Why it was not possible to describe the Archaeaon this primer pair.

Response: The 27F/1492R primers are universal primers designed for bacteria. They do not contain the conserved sequences of all archaea.

L336-342 missing information.

Response: Revised by adding more information.

Round 2

Reviewer 1 Report

I have no further comments.

Reviewer 2 Report

Dear authors!

Thank you for finalizing the results of your research!

Yours sincerely,

Reviewer